# GraphPINE: Graph importance propagation Neural Network for interpretable drug response prediction

## Abstract

Explainability is necessary for tasks that require a clear reason for a given result such as finance or biomedical research. Recent explainability methodologies have focused on attention, gradient, and Shapley value methods. These do not handle data with strong associated prior knowledge and fail to constrain explainability results by relationships that may exist between predictive features.

We propose a GraphPINE, a novel graph neural network (GNN) architecture that leverages domain-specific prior knowledge for node importance score initialization. Use cases in biomedicine necessitate generating hypotheses related to specific nodes. Commonly, there is a manual post-prediction step examining literature (i.e., prior knowledge) to better understand features. While node importance can be obtained for gradient and attention-based methods after prediction, these node importances lack complementary prior knowledge; GraphPINE seeks to overcome this limitation. GraphPINE differs from other GNNs with gating methods that utilize an LSTM-like sequential format such that we introduce an importance propagation layer that unifies 1) updates for feature matrix and node importances, jointly and 2) uses GNN-based graph propagation of feature values. This initialization and updating mechanism allows for more informed feature learning and improved graph representation.

We apply GraphPINE to cancer drug response prediction using pharmacogenomics data (i.e., both drug screening and gene data collected by several assays) for 5K gene nodes included in a gene-gene input graph with drug-target interaction (DTI) knowledge graph as initial importance. The gene-gene graph and DTIs were taken from literature curated prior knowledge sources and weighted by the literature information. GraphPINE demonstrates competitive performance and achieves a PR-AUC of 0.894 and ROC-AUC of 0.796 across 952 drugs. To highlight the interpretability aspect of our work, we provide the ability to generate sub-graphs of node importances. While our use case is related to biology, our work is generally applicable to tasks where information is separately known about feature relationships. Code: `https://anonymous.4open.science/r/GraphPINE-40DE`.

## 1 Introduction

Drug response prediction (DRP) is an open research challenge in personalized medicine and drug discovery. DRP and Drug-Target Interaction (DTI) represent distinct but interconnected concepts in pharmaceutical research. While DTI focuses on predicting the molecular interactions between drugs and specific proteins or genes, DRP aims to predict if biological systems (e.g., cells) are viable in response to drug treatments. There is no absolute requirement that DRP methods use biological interaction information. However, studies have shown value in the use of molecular interactions for DRP (Sokolov et al., 2016; Costello et al., 2014). Work in this research area seeks to improve treatment outcomes and reduce adverse effects. However, the complex interplay between drug compounds and cellular entities makes this task challenging. Traditional approaches often fail to capture the intricate network of interactions that influence drug response, leading to suboptimal predictions with limited interpretability. Despite recent advancements, current DRP methods still

face challenges such as data heterogeneity, limited sample sizes, and the need for multi-omics integration (Azuaje, 2017; Lu, 2018; Vamathevan et al., 2019).

Greater data availability combined with algorithmic improvements have led to an increase in machine learning (ML) techniques in this research area. GNNs have emerged as a promising approach due to their ability to model complex relational data (Kipf & Welling, 2016). Recent GNN variants, such as Graph Transformer Networks (Yun et al., 2019) and Graph Diffusion Networks (Klicpera et al., 2019), have shown promise in capturing complex, long-range dependencies in biological networks. However, these advanced architectures often come at the cost of increased complexity and reduced interpretability. This leads to two main limitations in existing GNN models for DRP. First, many models do not incorporate known biological information, such as DTI. This omission can lead to predictions that, while accurate, may not align with known biological mechanisms. Second, the "black box" nature of many deep learning models makes it difficult for researchers and clinicians to understand and trust the predictions. This lack of interpretability is a significant barrier to adopting these models for furthering understanding of drug mechanisms.

While some attempts have been made to incorporate biological priors into GNNs (Zitnik et al., 2018) or improve interpretability (Ying et al., 2019), no existing method addresses both challenges in the context of DRP. To address these limitations, we introduce GraphPINE (**Graph P**ropagating **I**mportance **N**etwork for **E**xplanation), a novel GNN approach combining the predictive power of deep learning with biologically informed feature importance propagation and interpretability.

GraphPINE builds upon recent advancements in GNNs, such as Graph Transformer (Shi et al., 2020) and explainable AI techniques (Ying et al., 2019). The key innovation of GraphPINE lies in its Importance Propagation (IP) Layer, which updates and propagates gene importance scores across the network during the learning process. This mechanism allows GraphPINE to:

1. Integrate known DTI information with the underlying gene network structure, ensuring the model's predictions are grounded in known biological interactions.

2. Capture drug-gene interactions with N-hops GNN layers, providing a more comprehensive view of drug influence on the gene network.

3. Generate interpretable visualizations of gene-gene interactions under the drug treatment, offering new perspectives on potential drug action mechanisms.

## 2 RELATED WORKS

The development of GraphPINE builds upon and extends several key areas of research in computational biology and ML. This section reviews relevant prior work in DRP, GNNs, and explainable AI for biological applications.

### 2.1 DRUG RESPONSE PREDICTION

Drug response prediction (DRP) refers to the process of forecasting how a particular drug will affect the vialbility of a biological system based on various data inputs such as genomic information and molecular structures (Adam et al., 2020; Fu et al., 2024). The goal is to predict the drug sensitivity, which can aid in personalized medicine, allowing for more targeted treatments for patients. The basic DRP can be described as the following function $y = f(G, D)$ where $y$ is the drug response value (e.g., IC50: half maximal inhibitory concentration, binary response: drug sensitive/resistance), $G$ is genomic features (e.g., gene expression, mutation), $D$ is drug features (e.g., molecular structure, chemical property information) and $f$ is an ML model.

Recently, Deep Learning (DL) has been applied to DRP, integrating multi-omics data. Several notable models have emerged: Li et al. (2019) developed DeepDSC which combines an autoencoder for gene expression to obtain hidden embeddings, which are then used as input to a feed-forward network along with drug fingerprint embeddings. Lao et al. (2024) implemented the DeepAEG, including transformer for SMILES and attention for multi-omics data (e.g., mutation, gene expression). Zhao et al. (2023) expands the Similarity Network Fusion (SNF) to the DRP, called Multi-source DRP (MSDRP). They combined multiple data sources (e.g., gene expression, mutation, SMILES, DTI) and used the SNF to fuse multiple information to make 2 interaction embeddings to obtain interaction information. A comprehensive review of this topic can be found in (Adam et al., 2020).

## 2.2 GRAPH NEURAL NETWORKS IN COMPUTATIONAL BIOLOGY

GNNs have emerged as a powerful tool for modeling complex biological systems. Their application to side effect prediction has demonstrated effectiveness in capturing drug-drug interactions (Huang et al., 2021). GNNs have also been used for molecular property prediction, showcasing the potential of GNNs in cheminformatics (Fu et al., 2021b). For the DRP, GraphDRP (Nguyen et al., 2021) integrates gene expression and protein-protein interaction networks, while MOFGCN (Peng et al., 2021) combines multi-omics data (e.g., gene expression).

## 2.3 EXPLAINABLE AI IN BIOLOGICAL APPLICATIONS

As ML models become complex, there is a growing need for interpretability, especially in biomedical applications where understanding the rationale behind predictions is fundamental for clinical research.

Explainable AI methods can be categorized into three main types:

1. **Gradient-based methods:** These techniques utilize gradient information to highlight important features. For example, Grad-CAM (Selvaraju et al., 2020) generates visual explanations for decisions made by convolutional neural networks. Fu et al. (2021a) produces molecular substructure-level gradient to provide interpretability for drug design.
2. **Attention-based methods:** These approaches leverage attention coefficients to identify relevant parts of the input. Abnar & Zuidema (2020) propose methods like attention rollout and attention flow to quantify the propagation of information through self-attention layers, improving the interpretability of Transformer models.

   For DRP, Inoue et al. (2024) employs Graph Attention Network (GAT) (Veličković et al., 2017) on a heterogeneous network of proteins, cell lines, and drugs, offering interpretability through attention coefficients. Shi et al. (2024) utilizes directed graph convolutional networks (GCN) (Kipf & Welling, 2016) to identify key features of drugs and cells in predicting therapeutic outcomes.
3. **Shapley value-based methods:** SHapley Additive exPlanations (SHAP) (Lundberg & Lee, 2017; Wang et al., 2024) assigns importance values to input features based on game theory principles, providing a unified measure of feature contributions to model predictions.

GraphPINE is most closely related to the attention-based methods, but with key distinctions. Unlike typical attention mechanisms that assign importance to edges, GraphPINE uses DTI information to initialize node importance scores. It propagates this importance throughout the learning process along with the graph structure. This approach incorporates prior biological knowledge, thereby enhancing biological interpretability.

## 2.4 INFORMATION PROPAGATION IN NEURAL NETWORKS

Earlier bioinformatic methods such as HotNet and Perturbation Biology were inspired by heat diffusion processes and message-passing algorithms (Leiserson et al., 2015), yet more recent work has focused on applying propagation concepts to neural network-based ML models.

Bach et al. (2015) introduced Layer-wise Relevance Propagation (LRP), a technique for explaining model decisions. LRP decomposes the prediction by propagating relevance scores from the output layer to the input layer, providing insights into each input feature's contribution to the final prediction.

Shrikumar et al. (2017) proposed DeepLIFT (Deep Learning Important FeaTures). This method computes importance scores, capturing non-linear dependencies that might be missed by other approaches. DeepLIFT addresses limitations of traditional gradient-based methods by considering the difference from a reference input. This approach offers a more nuanced understanding of feature contributions and provides more interpretable explanations of model outputs.

More recently, Abnar & Zuidema (2020) introduced Attention Flow, a method designed for Transformer models. This approach models the propagation of attention through the layers of a Transformer, quantifying how information flows from input tokens to output tokens. Attention Flow provides a more accurate measure of token relationships compared to raw attention weights, offering insights into how Transformer models process and utilize information across their multiple attention layers.

These methods can all be viewed as specialized forms of information propagation. In each case, the "information" being propagated represents the relevance, importance, or attention associated with

different components of the network. These approaches demonstrate how the concept of information propagation can be leveraged to enhance the interpretability of complex neural network models, offering valuable insights into their decision-making processes across various network architectures.

### 2.5 IMPORTANCE GATING WITH GNNS

Recent studies have proposed different approaches for incorporating gating mechanisms into GNNs. Two notable examples are Event Detection GCN (Lai et al., 2020) and CID-GCN (Zeng et al., 2021).

Event detection is a natural language processing (NLP) task that aims to identify specific events (such as accidents, or business transactions) from documents. Event Detection GCN implements a gating mechanism utilizing trigger candidate information (e.g., potential event-indicating words: "attacked", "acquired") to filter noise from hidden vectors. The model incorporates gate diversity across layers and leverages syntactic importance scores from dependency trees, which represent grammatical relationships between words in sentences.

CID-GCN, designed for chemical-disease relation extraction, constructs a heterogeneous graph with mentions (representing specific entity occurrences), sentences (containing the textual context), and entities (normalizing multiple mentions) nodes. The model employs gating mechanisms to address the over-smoothing problem in deep GCNs (Li et al., 2018) and enables effective information propagation between distant nodes.

GraphPINE advances these concepts through two key ideas. First, it introduces a novel approach to importance scoring by leveraging domain-specific prior knowledge for initialization, rather than relying solely on previous hidden states. Second, it implements a unified importance score updating mechanism through graph structure learning, departing from the context-based or two-step gating approaches of its predecessors.

## 3 METHODS

### 3.1 OVERVIEW OF GRAPHPINE

This section presents the GraphPINE model, including data preprocessing, network construction, and the model architecture. GraphPINE is a GNN architecture designed for accurate and interpretable DRP, leveraging multi-omics data such as gene expression, copy number variation, methylation, and mutation information, along with known biological interactions to provide comprehensive insights into drug-target relationships, as illustrated in Figure 1.

### 3.2 DATA PREPROCESSING AND NETWORK CONSTRUCTION

We collected multi-omics data (gene expression, methylation, mutation, and copy number variation) from NCI-60 cell lines, gene-gene interaction data from PathwayCommons, and DTI data from various sources including Comparative Toxicogenomics Database (CTD) (Davis et al., 2023), Drug-Bank (Wishart et al., 2018), the Drug Gene Interaction Database (DGIdb) (Freshour et al., 2021), the Search Tool for Interactions of Chemicals (STITCH) (Szklarczyk et al., 2021) and Kinase Inhibitor BioActivity (KIBA) dataset (Tang et al., 2014). For gene-gene network construction, we selected a subset of genes based on three criteria: variance in multi-omics data, network centrality, and DTI frequency. The final set consisted of 5,181 genes, forming a network with 630,632 interactions. Edge information among genes was categorized into seven types and encoded as one-hot vectors. The detailed explanation is in Appendix A.1.

In multi-omics data preprocessing, gene expression data was normalized through converting the data to the Transcripts Per Million (TPM), Log2 transformation, and winsorization. A 4-dimensional feature vector was created for each gene in each cell line, incorporating all multi-omics data.

For the DTI data, we integrated data from multiple sources (e.g., DrugBank and CTD) and calculated interaction scores based on available literature evidence. In this study, as a simplification, we will refer to both direct physical drug-target interactions (i.e., a chemical binding to a protein) as well as indirect interactions/associations (e.g., "drug results in increased acetylation of a protein" via intermediate events) as "DTI".

## A. Importance Propagation (IP) Layer

## B. GraphPINE architecture

## C. How to create the graph structure

**Figure 1: Overview of GraphPINE Components.** (A) Importance Propagation (IP) Layer: This illustrates the key components of the IP Layer in the GraphPINE model, including the GNN, importance gating, feature updates with residual connections, importance propagation, and updates. The symbols represent the following operations: $\sigma$ is the activation function, $\odot$ is element-wise multiplication, $\times$ is multiplication, $+$ is addition, $W$ denotes weighted calculation with bias, and $||$ represents concatenation. The parameter $\alpha$ is a hyperparameter for controlling importance. (B) GraphPINE architecture. (C) Data Creation Overview: The model integrates multi-omics data (gene expression, copy number, methylation, mutation) from NCI60 (Shoemaker, 2006) with gene-gene interaction networks from PathwayCommons (Cerami et al., 2010; Rodchenkov et al., 2019). Each edge has attributes such as "interact-with", which are converted into one-hot vectors for edge attribution.

Let $S_{dti}(d_i, g_j)$ be the initial importance score for drug $d_i$ and gene $g_j$. We normalized these scores to a range of [0.5, 1]:

$$\text{log\_count} = \log(1 + \text{PubMed ID\_count})$$

$$S_{dti}(d_i, g_j) = 0.5 + 0.5 \times \frac{\text{log\_count} - \min(\text{log\_count})}{\max(\text{log\_count}) - \min(\text{log\_count})}. \tag{1}$$

Here, log_count refers to the log-transformed PubMed ID counts, where PubMed ID_count represents the number of papers retrieved from PubMed ESearch(Sayers, 2009) using the query that searches for co-mentions by combining drug name and gene name. $d_i$ and $g_j$ denote specific drugs and genes. This

scaling process normalizes the data across different drug-gene pairs, facilitating comparative analysis and integration in steps. The 0.5 is added to distinguish the genes that are in databases, but they don't have the literature information. Therefore, the range of $S_{dti}(d_i, g_j)$ is $S_{dti}(d_i, g_j) \in \{0\} \cup [0.5, 1]$.

## 3.3 GRAPHPINE MODEL ARCHITECTURE

The GraphPINE model predicts drug response while learning and visualizing gene importance. It processes a gene interaction network $G = (V, E)$, node features $X \in \mathbb{R}^{|V| \times d}$, edge features $E_{attr} \in \mathbb{R}^{|E| \times f}$, and initial importance scores $I \in \mathbb{R}^{|V|}$, outputting a predicted drug response $y \in \mathbb{R}$ and updated gene importance scores $I' \in \mathbb{R}^{|V|}$. Given that our graph structure includes edge attributes, we explore GNN layers capable of handling edge attributes (i.e., Graph Attention Network (GAT) (Veličković et al., 2017), Graph Transformer (GT) (Yun et al., 2019), and Graph Isomorphism Network with Edge features (GINE) (Hu et al., 2019)).

### 3.3.1 IMPORTANCE PROPAGATION LAYER

The core component of GraphPINE is the Importance Propagation Layer (IP Layer), which operates in the following five main steps:

**1. GNN Layer** We apply a Graph Transformer (GT) layer (Yun et al., 2019) (TransformerConv) to process node features. For the node $i$, we have:

$$\mathbf{h}_i = \text{TransformerConv}(\mathbf{x}_i, \text{edge\_index}, \text{edge\_attr}), \tag{2}$$

where $\mathbf{h}_i$ is the output feature vector for node $i$, $\mathbf{x}$ represents input node features, edge_index denotes edge connections in the graph, and edge_attr represents edge attributes.

**2. Importance Gating** We generate a gate using the GT output and importance scores:

$$\mathbf{g}_i = \sigma(\mathbf{W}_g[\mathbf{h}_i \| I_i] + \mathbf{b}_g), \tag{3}$$

where $\mathbf{g}_i$ is the gate vector for node $i$, $\sigma$ is the sigmoid function, $\mathbf{W}_g$ and $\mathbf{b}_g$ are learnable parameters, $I_i$ is the importance score of node $i$, and $\|$ denotes concatenation. This gating mechanism, similar to LSTM/GRU gates (Hochreiter & Schmidhuber, 1997; Chung et al., 2014) but adapted for graphs, and also utilized with prior knowledge.

**3. Feature Matrix Update** We update node features using the generated gate:

$$\hat{\mathbf{x}}_i = \mathbf{g}_i \odot \mathbf{h}_i + (1 - \mathbf{g}_i) \odot \mathbf{x}_i, \tag{4}$$

where $\hat{\mathbf{x}}_i$ is the updated feature vector for node $i$, $\odot$ represents element-wise multiplication, and $\mathbf{x}_i$ is the original input feature vector for node $i$.

**4. Importance Propagation (IP)** We propagate importance scores through the network:

$$I'_i = \mathbf{W}_p \hat{\mathbf{x}}_i + b_p, \tag{5}$$

where $I'_i$ is the updated importance score for node $i$, and $\mathbf{W}_p$ and $b_p$ are learnable parameters.

**5. Importance Update** At each layer $l$, we update importance scores as follows:

$$I_i^{(l+1)} = \alpha I_i^{(l)} + (1 - \alpha) I_i'^{(l)}, \tag{6}$$

where $I_i^{(l)}$ is the importance score of node $i$ at layer $l$, $I_i'^{(l)}$ is the propagated importance score of node $i$ at layer $l$, and $\alpha$ is the importance decay rate.

Finally, we normalize and threshold the importance scores:

$$I_i^{\text{norm}} = \frac{I_i - \min(I)}{\max(I) - \min(I)}, \qquad I_i^{\text{final}} = \begin{cases} I_i^{\text{norm}} & \text{if } I_i^{\text{norm}} \geq \theta \\ 0 & \text{otherwise,} \end{cases} \tag{7}$$

where $I_i^{\text{norm}}$ is the normalized importance score for node $i$, $I_i^{\text{final}}$ is the final thresholded importance score for node $i$, and $\theta$ is the importance threshold.

**Insights into the IP Layer** The IP Layer offers several key advantages:

a) **Importance Score Propagation:** As defined in Equation 5, this mechanism learns a mapping from updated features to importance scores, considering both local and global graph structures.

b) **Adaptive Importance Update:** Equations 2 through 5 show how node features and importance scores are updated, balancing new and historical information for stability and adaptability.

c) **Computational Efficiency:** With a complexity of $O(|V| + |E|)$, IP Layer ensures scalability for large graphs.

d) **Enhanced Expressiveness:** GraphPINE captures indirect interactions through evolving importance scores (Equation 5), offering richer expressiveness than standard GNNs.

e) **Interpretability-Regularization Balance:** The gating mechanism (Equation 3) and feature update process (Equation 4) encourage sparse importance distributions, enhancing interpretability while maintaining model capacity.

These characteristics make the IP Layer a powerful tool for representation learning in graphs, suited for capturing complex drug-gene interactions while maintaining interpretability.

x

### 3.3.2 MODEL ARCHITECTURE

The GraphPINE model stacks three IP Layers, incorporating with GraphNorm for normalization, Dropout for regularization, and ReLU activation functions between layers. The final prediction is made by aggregating the node representations, applying a linear transformation, and then a sigmoid function to obtain the probability of the positive class (i.e., drug sensitivity) :

$$p = \sigma\bigg(\mathbf{W}_f\big(\frac{1}{|V|} \sum_{v \in V} \mathbf{h}_v^{(L)}\big) + b_f\bigg), \tag{8}$$

where $p$ is the probability of the positive class, $\mathbf{W}_f$ and $b_f$ are learnable parameters, $\mathbf{h}_v^{(L)}$ is the final node representation for node $v$ after $L$ layers, $|V|$ is the number of nodes, and $\sigma$ is the sigmoid function. The output $p$ represents the probability of the positive class. The learning objective is a combination of binary cross entropy (BCE) and an importance regularization term to encourage sparsity in gene importance scores. The total loss function is as follows:

$$\mathcal{L} = \mathcal{L}_{\text{BCE}} + w_{\text{imp}} \cdot \mathcal{L}_{\text{imp}}, \tag{9}$$

where $\mathcal{L}_{\text{BCE}}$ is the binary cross entropy between predicted and actual drug responses, $\mathcal{L}_{\text{imp}}$ is the L1 regularization term on the importance scores. $w_{\text{imp}}$ is weighting hyperparameter.

## 4 EXPERIMENTS

### 4.1 DATASET

We processed the IC50 data from the NCI-60 dataset (Shoemaker, 2006) using the rcellminer (Luna et al., 2016) package to create a binary classification. We applied an empirically determined threshold of -4.595 to the log-transformed IC50 values, initially resulting in an equal distribution of labels. Subsequently, we selected only the drugs that were present in multi-omics datasets. This selection process led to an imbalanced dataset, with the resulting dataset consisting of 53,852 entries in total, comprising 36,171 positive and 17,681 negative instances.

We established a zero-shot prediction scenario by randomly allocating 70% of cell lines and 60% of NSC identifiers to the training and validation sets (571 drugs and 42 cell lines), with the remaining forming the test set (381 drugs and 18 cell lines). This ensures no overlap between train/validation and test sets, allowing us to evaluate the model's ability to generalize to unseen drug-cell line pairs.

The data was split into 18,067 training, 4,516 validation, and 6,525 test entries. Class 0 represents drug sensitive, while class 1 indicates drug resistance. The training set has 5,892 sensitive and 12,175 resistant cases. The validation set contains 1,487 sensitive and 3,029 resistant cases. The test set includes 2,154 sensitive and 4,371 resistant cases. Detailed procedures can be found in Appendix B.

## 4.2 Prediction Performance

To evaluate the effectiveness of GraphPINE, we conducted a comparison against several baseline methods, including 5 traditional ML approaches, 2 current research methods, and 3 GNNs without IP layer. Table 1 presents the performance metrics for each method, averaged over 5 independent runs.

| | Methods | Explainability | ROC-AUC (↑) | PR-AUC (↑) | Accuracy (↑) | Precision (↑) | Specificity (↑) |
|---|---|---|---|---|---|---|---|
| **Baseline** | RF | Feature Importance | 0.7877 (±0.0011) | 0.8917 (±0.0016) | 0.7164 (±0.0021) | 0.7263 (±0.0020) | 0.6320 (±0.0034) |
| | LightGBM | Feature Importance | 0.7901 (±0.0000) | 0.8697 (±0.0000) | 0.7465 (±0.0000) | 0.7686 (±0.0000) | 0.4568 (±0.0000) |
| | MLP | - | 0.7498 (±0.0098) | 0.8384 (±0.0059) | 0.7100 (±0.0036) | 0.7208 (±0.0092) | 0.2706 (±0.0510) |
| | MPNN | - | 0.7920 (±0.0125) | 0.8924 (±0.0057) | 0.7276 (±0.0086) | 0.7257 (±0.0114) | 0.5709 (±0.0444) |
| | GCN | - | 0.7660 (±0.0185) | 0.8715 (±0.0101) | 0.7096 (±0.0207) | 0.7096 (±0.0199) | 0.5593 (±0.0288) |
| **Previous Research** | DeepDSC | - | 0.7127 (±0.0135) | 0.7833 (±0.0085) | **0.7514** (±0.0112) | **0.8071** (±0.0092) | 0.5990 (±0.0210) |
| | MOFGCN | - | 0.4922 (±0.0000) | 0.6660 (±0.0000) | 0.3546 (±0.0000) | 0.6495 (±0.0000) | **0.9006** (±0.0000) |
| **Ablation w/o IP layer** | GAT | - | 0.7580 (±0.0190) | 0.8682 (±0.0133) | 0.7031 (±0.0070) | 0.6867 (±0.0111) | 0.3953 (±0.0656) |
| | GT | - | 0.7743 (±0.0188) | 0.8739 (±0.0163) | 0.7173 (±0.0197) | 0.7167 (±0.0193) | 0.5664 (±0.0336) |
| | GINE | - | 0.7501 (±0.0193) | 0.7936 (±0.0141) | 0.6999 (±0.0177) | 0.6696 (±0.0177) | 0.3360 (±0.0372) |
| **GraphPINE** | GAT | Node Importance | 0.7886 (±0.0057) | 0.8920 (±0.0047) | 0.7196 (±0.0123) | 0.7165 (±0.0130) | 0.5472 (±0.0479) |
| | GT | Node Importance | **0.7955** (±0.0055) | **0.8939** (±0.0013) | 0.7235 (±0.0051) | 0.7192 (±0.0057) | 0.5478 (±0.0189) |
| | GINE | Node Importance | 0.7903 (±0.0032) | 0.8911 (±0.0013) | 0.7298 (±0.0121) | 0.7280 (±0.0154) | 0.5749 (±0.0560) |

**Table 1: Predictive Performance Comparison for Binary Classification.** Results show averages of 5 independent runs with standard deviations in parentheses. Best values for each metric are in **bold**. Abbreviations: ROC-AUC: Receiver Operating Characteristic Area Under the Curve, PR-AUC: Precision-Recall Area Under the Curve, RF: Random Forest, MLP: Multiple Layer Perceptron, MPNN: Message-Passing Neural Network, GCN: Graph Convolutional Networks, MOFGCN: Multi-Omics Data Fusion and Graph Convolution Network, GAT: Graph Attention Network, GT: Graph Transformer, GINE: Graph Isomorphism Network with Edge features. Feature Importance: A measure of how much each feature contributes to a model's predictions.

Our proposed GraphPINE model, particularly the Graph Transformer (GT) variant, demonstrates superior performance across multiple key metrics. Given the imbalanced nature of our dataset, which is common in biological interaction prediction tasks, we place particular emphasis on the PR-AUC and ROC-AUC scores as the most critical evaluation metrics.

Notably, GraphPINE (GT) achieves the highest PR-AUC (0.8939) and ROC-AUC (0.7955), underscoring its effectiveness in handling imbalanced data. While DeepDSC shows higher accuracy (0.7514) and precision (0.8071), GraphPINE (GT)'s balanced performance across multiple metrics, particularly in PR-AUC and ROC-AUC, indicates its robust ability to effectively discriminate between classes while maintaining a strong balance between precision and recall.

MOFGCN exhibits a performance pattern with a high specificity (0.9006) but poor performance across other metrics (ROC-AUC: 0.4922, PR-AUC: 0.6660, Accuracy: 0.3546). This suggests that while the model excels at identifying resistance, it does so at the expense of overall classification performance, indicating a highly imbalanced prediction behavior that limits its practical utility.

The ablation study demonstrates the significant impact of the IP layer across all architectures. The GT variant achieves the best performance with PR-AUC of 0.8939 and ROC-AUC of 0.7955, representing improvements of 2.29% and 2.74% from its baseline scores of 0.8739 and 0.7743, respectively. The GAT architecture exhibits notable enhancements with PR-AUC increasing by 2.74% (from 0.8682 to 0.8920) and ROC-AUC by 4.04% (from 0.7580 to 0.7886). Most remarkably, the GINE architecture shows the most substantial improvement, with PR-AUC increasing by 12.29% (from 0.7936 to 0.8911) and ROC-AUC by 5.36% (from 0.7501 to 0.7903), demonstrating the IP layer's effectiveness in enhancing model performance.

It is worth noting that all variants of GraphPINE (GINE, GAT, and GT) show low standard deviations across runs, indicating the stability and reliability of our proposed method. This consistency is

valuable when dealing with imbalanced datasets, as it suggests that our model's performance is robust across different data splits and initializations.

## 4.3 INTERPRETABILITY ANALYSIS

9-Methoxycamptothecin-Related Gene Interaction Network

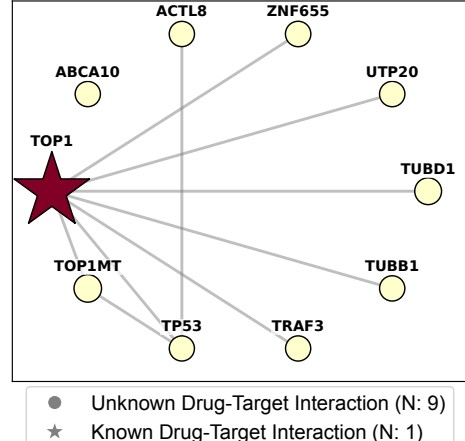

| Rank | Initial Importance | Gene | PMIDs | Relationship |
|------|--------------------|----------|----------|--------------|
| 1 | 1 | TOP1 | 29312794... | Target |
| 2 | - | TOP1MT | 24890608... | Indirect |
| 3 | - | TUBD1 | - | - |
| 4 | - | ZNF655 | - | - |
| 5 | - | UTP20 | - | - |
| 6 | - | TUBB1 | - | - |
| 7 | - | ACTL8 | - | - |
| 8 | - | ABCA10 | 10606239 | Indirect |
| 9 | - | TRAF3 | - | - |
| 10 | - | TP53 | 12082016... | Indirect |

Table 2: **Top 10 predicted important genes for 9-Methoxycamptothecin and related literature.** (-) represents no initial DTI (0) and (...) describes multiple papers. Target: Genes encoding proteins that directly bind to and interact with the drug. Indirect: Genes that do not encode proteins that physically interact with the drug but are involved in its mechanism of action, pathway, or response.

**Figure 2: Gene importance scores for 9-Methoxycamptothecin.** Node size describes the propagated gene importances and node color shows the initial DTI score.

GraphPINE assigns importance scores to each gene, indicating their relative significance in predicting drug responses. Figure 2 illustrates the gene interaction network associated with 9-Methoxycamptothecin, a DNA damage-related anticancer drug and derivative of camptothecin. In this network, the size of each node reflects the propagated gene importance after prediction, while the node shape differentiates between known DTIs (denoted by a star) and unknown interaction partners (denoted by a circle). The color of the nodes represents known DTI scores. The known target of 9-Methoxycamptothecin is TOP1; other genes are potential but not confirmed interactors. Figure 2 shows that the known target, TOP1, has the highest DTI score and propagated importance, and other genes have propagated importance but are low compared with TOP1. ABCA10 lacks an edge because it is not among the top interactions shown. Expanding the graph could reveal its connections, as it might interact with other genes beyond the top 10 displayed.

Table 2 lists the top 10 important genes related to 9-Methoxycamptothecin The highest-scoring gene, TOP1, is a known target of 9-Methoxycamptothecin. Although TOP1MT is not known as a target of 9-Methoxycamptothecin, it is a target of camptothecin, suggesting that 9-Methoxycamptothecin may also target TOP1MT. Additionally, there is an established association between camptothecin and ABC transporters, making it plausible that ABCA10 might also be related to 9-Methoxycamptothecin. Moreover, the efficacy of 9-Methoxycamptothecin, a TOP1 inhibitor, may be influenced by the status of TP53, which modulates cellular responses to DNA damage.

These results show that GraphPINE can obtain some biological relationship from gene-gene networks with prior DTI information. TOP1MT and TP53 are already known as related genes (Chen et al., 2021) but not known as DTI, and our model successfully detects this relationship.

## 4.4 EVALUATION OF IMPORTANCE SCORE PROPAGATION

To understand the extent to which our importance propagation affects our initial importance scores, we analyzed 6000 randomly selected drug-cell combinations (389 unique drugs × 26 cell lines) across 5181 genes. Our prior knowledge interaction data is highly sparse, each drug was associated with between 1 and 956 interactors, with an average of 39.86 interactions; Appendix B.4 includes a distribution of the number of interactions (Table 3). Importance scores of 0 imply the absence of an interaction, and non-zero values imply an interaction. Therefore, we first examine the extent that our

propagation method increased non-zero values. We observed that non-zero values increased from 0.77% to 39.8% after propagation; this increased the average number of non-zero values per drug from 39.86 to 2061.81 (Appendix B.4). Next, we examined how much individual non-zero values were altered by propagation using a similarity comparison and, also, a rank change analysis. For the similarity analysis (using cosine similarity and Spearman rank correlation), we observe a high, but not perfect correlation (0.89 and 0.82, respectively); this suggests importance values that are updated as part of the training process. Approximately, 90% of importance values showed some rank change as an effect of propagation with an average shift of ±67.02 (maximum +946/-932), Next, we considered the situation of starting with random initial importance values, and asked if training shifts these values toward our prior knowledge-derived importance values.

## 5 DISCUSSION

We introduced GraphPINE, a novel interpretable GNN architecture featuring an "Importance Propagation Layer". This architecture allows us to highlight node importance under stringent constraints, incorporating prior knowledge such as biological or medicinal information.

While our study focuses on DRP, the Graph-PINE framework holds potential for a wide range of applications in fields that involve complex network structures with inherent node importance. For instance, PageRank scores could be used as initial importance values to enhance the propagation of relevance among web pages in web graph analysis. In traffic network optimization, the usage frequency of stations or intersections could serve as initial importance to improve traffic flow efficiency. These examples showcase GraphPINE's versatility and open opportunities for validating the effectiveness of importance propagation mechanisms across diverse domains.

| Metric | Value |
|---|---|
| Cosine sim. | 0.87 |
| Spearman corr. | 0.82 |
| Rank changes | 90.42% |
| Avg. shift | ±67.02 |
| Max up | 946 |
| Max down | -932 |

Table 3: **Differences in Node (Gene) Ranks Before and After Propagation.** Cosine sim.: Cosine similarity between initial/propagated importance rank. Spearman corr.: Spearman Rank correlation between initial/propagated importance rank. Rank changes: The percentage of genes whose ranks changed after propagation. Avg. shift: The average rank shift. Max up/down: Maximum upward/downward rank mobility.

Importantly, this model is designed to directly predict whether a drug will be effective for a patient. GraphPINE may have potential applications in drug discovery and personalized medicine research. It could assist in predicting drug responses in preclinical stages, potentially informing the selection of promising candidates for further study. For instance, it could be a useful tool in rare disease research where patient data is scarce. In personalized medicine research, GraphPINE's interpretable predictions could provide hypothesis for researchers studying treatment responses.

One limitation of our research is that the interpretability of the model relies on the graph structure, and the importance propagation is graph-based, despite the inclusion of initial information through residual connections. This reliance can sometimes be too restrictive, necessitating the integration of additional information. Therefore, it may be beneficial to explore the combination of graph-based propagation with other knowledge sources, such as protein-protein interactions or alternative network structures, to enhance the model's accuracy and interpretability. In addition, this model heavily relies on the quality of the DTI dataset. While we merged several datasets, the overall quality is not yet satisfactory. It is one of our ongoing goals to improve the coverage of this data to improve the model's performance.

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

# A IMPLEMENTATION DETAILS AND HYPERPARAMETER TUNING

## A.1 DATA PREPROCESSING AND NETWORK CONSTRUCTION

We integrated multiple data sources to create a comprehensive gene-gene interaction network and DTI dataset. Our approach involves several key steps.

### A.1.1 DATA INTEGRATION

Let $G = g_1, g_2, ..., g_n$ be the set of all genes, and $D = d_1, d_2, ..., d_m$ be the set of all drugs. We collected data from various sources. From NCI-60 cell lines, we obtained multi-omics data including gene expression ($X_{exp} \in \mathbb{R}^{n \times c}$), methylation ($X_{met} \in \mathbb{R}^{n \times c}$), mutation ($X_{mut} \in 0, 1^{n \times c}$), and copy number variation (CNV) ($X_{cnv} \in \mathbb{R}^{n \times c}$), where $n$ is the number of genes and $c$ is the number of cell lines. Gene-gene interaction data ($E_{gg} \subseteq G \times G$) was sourced from PathwayCommons (Cerami et al., 2010; Rodchenkov et al., 2019), containing various types of interactions such as catalysis-precedes, controls-expression-of, controls-phosphorylation-of, controls-state-change-of, controls-transport-of, in-complex-with, and interacts-with. DTI data ($E_{dti} \subseteq D \times G$) was collected from multiple sources, including the CTD, DrugBank, DGIdb, STITCH, and the KIBA dataset.

### A.1.2 GENE-GENE NETWORK CONSTRUCTION

We selected a subset of genes $G' \subseteq G$ based on three criteria. (1) First, we considered variance in multi-omics data. For each data source $s \in \{exp, met, mut, cnv\}$ where exp represents gene expression, met represents methylation, mut represents mutation, and cnv represents copy number variation, we computed the variance for each gene across cell lines:

$$\text{var}_s(g_i) = \frac{1}{c-1} \sum_{j=1}^{c} (X_{s_{ij}} - \bar{X}s_i)^2 \tag{10}$$

We selected the top 3000 genes with the highest variance for each data source. (2) Second, we computed network centrality, calculating the degree centrality for each gene in the initial interaction network:

$$\text{centrality}(g_i) = \frac{|(g_i, g_j) \in E_{gg} \vee (g_j, g_i) \in E_{gg}|}{|G| - 1} \tag{11}$$

We selected the top 3000 genes with the highest centrality. (3) Third, we considered DTI frequency, calculating the frequency of each gene in the DTI data:

$$\text{freq}_{\text{dti}}(g_i) = |(d_j, g_i) \in E_{dti}| \tag{12}$$

We selected the top 3000 genes with the highest DTI frequency. The final set of genes $G'$ was the union of these selections, resulting in 5,181 genes. We then constructed the gene-gene interaction network $G' = (V', E')$, where $V' = G'$ and $E' = E_{gg} \cap (G' \times G')$, containing 630,632 interactions.

### A.1.3 EDGE ENCODING

Each interaction between genes is categorized into one of seven types based on the information from PathwayCommons:"catalysis-precedes", "controls-expression-of", "controls-phosphorylation-of", "controls-state-change-of", "controls-transport-of", "in-complex-with", and "interacts-with". These interaction types were encoded as one-hot vectors.

Let $T = \{t_1, t_2, ..., t_7\}$ represent the set of all interaction types. For each edge $e \in E'$, a binary vector $v_e \in \{0, 1\}^7$ was created, where each element corresponds to a specific interaction type:

$$v_e[i] = \begin{cases} 1 & \text{if edge } e \text{ has interaction type } t_i \\ 0 & \text{otherwise.} \end{cases} \tag{13}$$

### A.1.4 Multi-omics Data Preprocessing

We focused on normalizing gene expression data through several steps. First, we converted the data to Transcripts Per Million (TPM):

$$\text{TPM}_{ij} = \frac{X_{\exp_{ij}}}{\sum_{i=1}^{n} X_{\exp_{ij}}} \times 10^6. \tag{14}$$

Next, we applied a Log2 transformation:

$$X'_{\exp ij} = \log_2(\text{TPM}_{ij} + 1). \tag{15}$$

Finally, we performed Winsorization. Let $q_{0.1}$ and $q_{99.9}$ be the 0.1 and 99.9 percentiles of $X'exp$. We applied:

$$X''_{\exp_{ij}} = \begin{cases} q_{0.1} & \text{if } X'_{\exp_{ij}} < q_{0.1} \\ q_{99.9} & \text{if } X'_{\exp_{ij}} > q_{99.9} \\ X'_{\exp_{ij}} & \text{otherwise.} \end{cases} \tag{16}$$

These steps ensured our gene expression data was normalized and scaled for further analysis. We then created 4-dimensional feature vectors for each gene in each cell line:

$$X_i = \left[ X''_{\exp_i}, X_{\text{met}_i}, X_{\text{mut}_i}, X_{\text{cnv}_i} \right]. \tag{17}$$

### A.2 Implementation Details

The GraphPINE model was implemented using Python 3.10, PyTorch 2.4.0 and PyTorch Geometric 2.5.3, leveraging their efficient deep learning and graph processing capabilities. We employed the Adam optimizer for training, with a learning rate of 0.001 and a batch size of 32. The model architecture incorporates 3 Importance Propagation Layers ($L = 3$), each containing 64 hidden units. To balance model performance and interpretability, we set the importance regularization coefficient $\lambda$ to 0.01 and the importance threshold $\tau$ to 0.1.

All experiments were conducted on NVIDIA Tesla A100 GPUs with 80 GB memory. The average training time for GraphPINE was 0.2 seconds, with an inference time of 0.1 seconds per drug-cell line pair, demonstrating its feasibility for large-scale DRP tasks. To ensure reproducibility and facilitate further research, we have made our code and datasets publicly available at `https://anonymous.4open.science/r/GraphPINE-40DE`.

### A.3 Training Procedure

The training procedure for the GraphPINE model is designed to optimize performance while preventing overfitting. Algorithm 1 presents a detailed overview of this process. Concretely, GraphPINE training procedure involves initializing model parameters, iterating through epochs, performing forward and backward passes, computing losses, and updating parameters. The procedure also includes an early stopping mechanism to prevent overfitting.

We employ the Adam optimizer with an initial learning rate of $\eta = 10^{-3}$.

### A.4 Hyperparameter Tuning

To optimize the performance of our GraphPINE model, we conducted extensive hyperparameter tuning using Optuna (Akiba et al., 2019), an efficient hyperparameter optimization framework. We utilized MLflow for experiment tracking and logging, ensuring comprehensive documentation of our optimization process.

Our hyperparameter search space encompassed key model parameters, including the number of epochs (1-3), number of attention heads (1, 2, 4), number of GNN layers (2-4), dropout rate (0.1-0.3), importance decay (0.7-0.9), importance threshold (1e-5 to 1e-3), hidden channel size (16, 32), BCE weight (0.9-1.1), importance regularization weight (0.005-0.02), and learning rate (0.001-0.1). The

---

**Algorithm 1** GraphPINE Training Procedure

---

1: Initialize model parameters $\theta$
2: Initialize optimizer with learning rate $\eta$
3: Set early stopping patience $p$ and minimum delta $\delta$
4: **for** epoch = 1 to $T_{\text{total}}$ **do**
5:     **for** batch in training data **do**
6:         Forward pass: $\hat{y}, I' = f_\theta(X, E, I)$
7:         Compute loss: $L = w_{\text{BCE}} \cdot \mathcal{L}_{\text{BCE}}(\hat{y}, y) + w_{\text{imp}} \cdot \mathcal{L}_{\text{imp}}(I', I)$
8:         Backward pass: Compute $\bullet_\theta \mathcal{L}$
9:         Update parameter using Adam optimizer.
10:     **end for**
11:     Evaluate on validation set
12:     **if** validation loss improved by at least $\delta$ **then**
13:         Reset patience counter
14:         Save best model
15:     **else**
16:         Decrement patience counter
17:         **if** patience counter = 0 **then**
18:             Early stop and return best model
19:         **end if**
20:     **end if**
21: **end for**

---

batch size was initially set to 5, with a dynamic reduction mechanism implemented to handle potential memory constraints.

The optimization process consisted of 20 trials, each involving the following steps: (1) hyperparameter suggestion by Optuna, (2) GraphPINE model initialization with the suggested configuration, (3) model training and validation, and (4) reporting of the minimum validation loss as the objective value for optimization. This systematic approach allowed us to identify the optimal hyperparameter configuration that balanced model performance and computational efficiency.

Throughout the implementation and tuning process, we leveraged several key libraries and tools. PyTorch served as the foundation for building and training our neural network model. Optuna facilitated efficient hyperparameter optimization, while MLflow provided robust experiment tracking and logging capabilities. We also utilized NumPy for numerical computations and Pandas for data manipulation and analysis, ensuring a comprehensive and efficient development environment.

This rigorous implementation and tuning process enabled us to develop a highly optimized Graph-PINE model, capable of accurate and interpretable DRPs. The combination of advanced deep learning techniques, efficient hyperparameter optimization, and careful implementation considerations resulted in a model that balances performance, interpretability, and computational efficiency.

A.5  BASELINE SETTING

We implemented three baseline models for comparison: Random Forest (RF), LightGBM, and Multiple Layer Perceptron (MLP). All models were trained on the same dataset, which combined gene expression, methylation, mutation, copy number variation, and drug-target interaction data.

**Random Forest (RF):**  We used `scikit-learn`'s `RandomForestClassifier` with hyperparameters optimized via `Optuna`. The key hyperparameters included the number of estimators (100–1000), max depth (10–100), min samples split (2–20), min samples leaf (1–10), and max features (None, `"sqrt"`, or `"log2"`).

**LightGBM:**  We implemented LightGBM (Ke et al., 2017) with binary classification objective and log loss metric. Hyperparameters were tuned using `Optuna`, including num_leaves (31–255), learning_rate (1e-3 to 1.0), feature_fraction (0.1–1.0), bagging_fraction (0.1–1.0), bagging_freq (1–

7), min_child_samples (5–100), lambda_l1 and lambda_l2 (1e-8 to 10.0), and num_boost_round (100–2000).

**Multiple Layer Perceptron (MLP):**  We created a `PyTorch`-based MLP with a flexible archi-tecture. Hyperparameters optimized via `Optuna` included the number of layers (2–5), hidden dimensions (64–512 units per layer), learning rate (1e-5 to 1e-1), batch size (32, 64, 128, or 256), dropout rate (0.1–0.5), and normalization type (batch or layer normalization).

**DeepDSC and MOFGCN:**  For DeepDSC, we follow the original architecture consisting of a stacked autoencoder followed by a feed forward network. The encoder comprises three hidden layers (2,000, 1,000, and 500 units) while the decoder mirrors this with hidden layers of 1,000 and 2,000 units. The activation function is selu for hidden layers and sigmoid for the output layer. Training employs AdaMax optimizer with learning rate 0.0001, gradient clipping at 1.0, and Xavier uniform initialization1.

For MOFGCN, we utilize the following hyperparameters: scale parameter $\varepsilon = 2$, proximity parameter $N = 11$, number of iterations $t = 3$, embedding dimension $h = 192$, correlation information dimension $k = 36$, scaling parameter $\alpha = 5.74$, learning rate $5 \times 10^{-4}$, and 1000 training epochs. The model uses PyTorch framework with Adam optimizer.

Both models employ early stopping to prevent overfitting - DeepDSC with patience of 30 epochs and MOFGCN monitoring the validation loss.

**MPNN, GCN, and GINE:**  For the MPNN (Message-Passing Neural Network) (Gilmer et al., 2017), GCN (Graph Convolutional Network) (Kipf & Welling, 2016), and GINE (Graph Isomorphism Network with Edge features) (Hu et al., 2019), we tuned the hyperparameters using the following configuration. The number of epochs (num_epochs) was selected from {10, 50, 100}. The batch size was chosen from {2, 3, 4}. The number of GNN layers was selected from {1, 2, 3}. The dropout rate was selected from {0.1, 0.2, 0.3}. The importance decay was chosen from {0.7, 0.8, 0.9}. The importance threshold was selected from {1e-5, 1e-4, 1e-3}. The hidden channel size was selected from {16, 32}. The weight for the mean squared error loss was selected from {0.9, 1.0, 1.1}. The weight for importance regularization was selected from {0.005, 0.01, 0.02}. The learning rate was selected from {0.001, 0.01, 0.1}.

**GAT and Graph Transformer:**  For the GAT (Graph Attention Network) (Veličković et al., 2017) and Graph Transformer models (Yun et al., 2019), we used a similar hyperparameter tuning configuration as for MPNN, GCN, and GINE. However, for GAT and Graph Transformer, we also included the number of attention heads, which was selected from {1, 2, 4}. This additional parameter helps in controlling the number of attention mechanisms in the model, enabling it to learn more complex representations.

For all models, we used `Optuna` for hyperparameter optimization, maximizing accuracy on the validation set. Each model was then trained five times with the best hyperparameters, and we report the mean and standard deviation of accuracy, precision, recall, and F1 score on the test set.

The data preprocessing steps were consistent across all models, including normalization of gene expression data and concatenation of multi-omics features. This ensured a fair comparison between the baseline models and our proposed GraphPINE method.

# B  EXPERIMENTS

## B.1  DATASET AND PREPROCESSING

In this study, we utilized a comprehensive drug response dataset containing information on multiple cell lines and compounds. The dataset was preprocessed and split to ensure a rigorous evaluation of the model's generalization capabilities. Initially, the dataset contained IC50 data for unique cell lines and unique NSC (Cancer Chemotherapy National Service Center number) identifiers for compounds.

To adapt this data for binary classification, we applied an empirically determined threshold, which was set to achieve an approximately 50:50 ratio of response to non-response using below formula.

$$\text{binarize}(x) = \begin{cases} 1 & \text{if } x < \text{ threshold} \\ 0 & \text{otherwise,} \end{cases}, \tag{18}$$

where threshold is the hyperparameter and we set -4.595.

This process resulted in a dataset of 331,558 entries. We then refined our dataset to focus on the 60 cell lines present in the NCI60 panel, reducing the data to 315,778 entries. Further narrowing our scope to include only the drugs used in the NCI60 project, we arrived at a final dataset of 53,852 entries.

To set up a zero-shot prediction scenario, we randomly selected 70% of unique cell lines and 60% of unique NSC identifiers for the training and validation sets. The remaining cell lines and NSC identifiers were used for the test set, ensuring no overlap of cell lines or compounds between the train/validation and test sets. This approach allows us to evaluate the model's ability to generalize to entirely new cell-compound combinations.

The data was split as follows: The training set comprises 18,067 entries, consisting of 571 unique drugs (NSCs) and 42 unique cell lines. The validation set contains 4,516 entries, utilizing the same 571 drugs and 42 cell lines as the training set. The test set includes 6,525 entries, encompassing 381 unique drugs and 18 unique cell lines.

Notably, while the training and validation sets share common cell lines and drugs, the test set introduces novel drug-cell line combinations. This configuration allows for a rigorous assessment of our model's generalization capability, enabling us to evaluate its predictive performance on unseen drug-cell line pairs.

### B.2 EVALUATION METRICS

We evaluated GraphPINE using a comprehensive set of metrics to assess its classification performance. The Accuracy was used to measure the overall correctness of the model's predictions across all classes. To provide a more nuanced assessment of the model's discriminative ability, we calculated the Area Under the Receiver Operating Characteristic curve (ROC-AUC) and the Area Under the Precision-Recall curve (PR-AUC). ROC-AUC quantifies the model's ability to distinguish between classes across various threshold settings, while PR-AUC is particularly useful for evaluating performance on imbalanced datasets. To further characterize the model's performance on negative instances, we computed the Specificity, which measures the proportion of actual negatives correctly identified. Additionally, we calculated the Negative Predictive Value (NPV), which quantifies the proportion of negative predictions that were actually correct. These metrics collectively offer a thorough evaluation of GraphPINE's ability to correctly classify both positive and negative instances, providing insights into its performance across different aspects of the classification task.

### B.3 INTERPRETABILITY ANALYSIS

Figure 3 shows the predicted interaction network for a roscovitine derivative. The network contains mostly unknown interactions (9) with only one known interaction. CDK1 is highlighted as the most important predicted target gene. This suggests the roscovitine derivative may have novel mechanisms of action beyond the known CDK inhibition, but CDK1 remains a key target.

Table 4 lists the top 10 predicted important genes for the roscovitine derivative. CDK1 is ranked first, consistent with roscovitine's known mechanism as a CDK inhibitor. However, most other predicted genes like NDE1, INCENP, EEF1D etc. are novel interactions without existing evidence. This suggests potential new pathways the derivative may affect beyond CDK inhibition.

Figure 4 displays a heatmap illustrating the drug-gene co-occurrences based on PubMed abstracts. Out of 3,810 drug-gene propagated importance-based relationships, 464 of the drug-gene co-occurrences were found in the abstracts. Figure 4 shows the subset of the heatmap. This subset displays only instances where each drug has DTIs under 10. The color corresponds to the number of publications identified by log scale. Similarly, the symbols in Figure 4 indicate different relationship types. The ♥ symbol represents relationships predicted by GraphPINE and present in the DTI dataset (61 instances

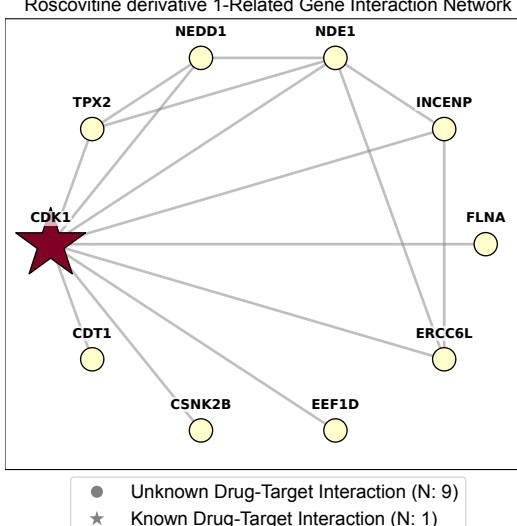

**Figure 3:** Gene importance scores and interactions for Roscovitine derivative 1. Node size describes the propagated gene importances.

**Table 4:** Top 10 predicted important genes for Roscovitine derivative 1.

| Rank | Gene Name | Evidence (PMID) |
|------|-----------|-----------------|
| 1 | CDK1 | 37635245 |
| 2 | NDE1 | - |
| 3 | INCENP | - |
| 4 | EEF1D | - |
| 5 | NEDD1 | - |
| 6 | CDT1 | 35931300 |
| 7 | CSNK2B | - |
| 8 | TPX2 | - |
| 9 | ERCC6L | - |
| 10 | FLNA | - |

in the subset). The ♣ symbol indicates relationships predicted by GraphPINE but not present in the DTI dataset (20 instances in the subset). The ♦ symbol denotes relationships only present in the DTI dataset (8 instances in the subset). This demonstrates our model's potential to capture known drug-target information and suggest new drug-gene relationships.

## B.4 EVALUATION OF IMPORTANCE SCORE PROPAGATION

To validate our importance propagation mechanism's effectiveness, we analyzed rank comparisons before/after propagation across 6000 randomly selected drug-cell combinations (389 unique drugs, 26 unique cell lines) and 5181 genes.

The initial importance density was 0.77% with an average of 39.86 interactions per drug-cell combination. After propagation, the interactions density increased to 39.8% (+38.96%) with 2061.81 average interactions.

The metrics comparison revealed a high cosine similarity of 0.9, indicating that 90% of genes maintained their original characteristics post-propagation. While the overall Spearman rank correlation was low due to zero entries, non-zero entries showed a strong correlation of 0.81, confirming preservation of meaningful relationships during network expansion.

99.98% of genes showed rank changes, with an average shift of ±1156.82 positions (maximum: +2658, minimum: -2590). This substantial change, combined with high similarity (0.90) to original data, indicates successful discovery of hidden connections while maintaining data integrity.

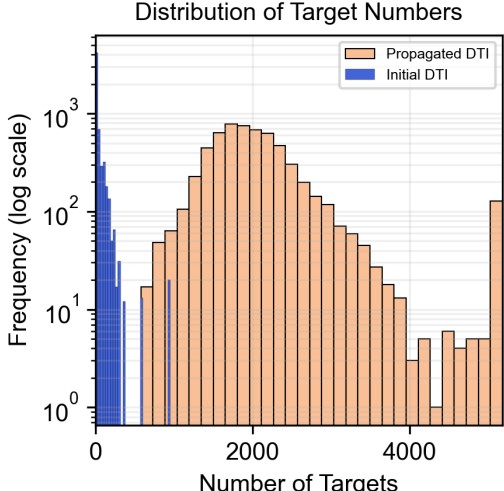

**Figure 5:** Distribution of Interactions Numbers Before/After Propagation. Initial interactions (blue) shows a concentrated distribution near zero interactions, while Propagated interactions (orange) demonstrates a broader distribution centered around 2000 interactions.

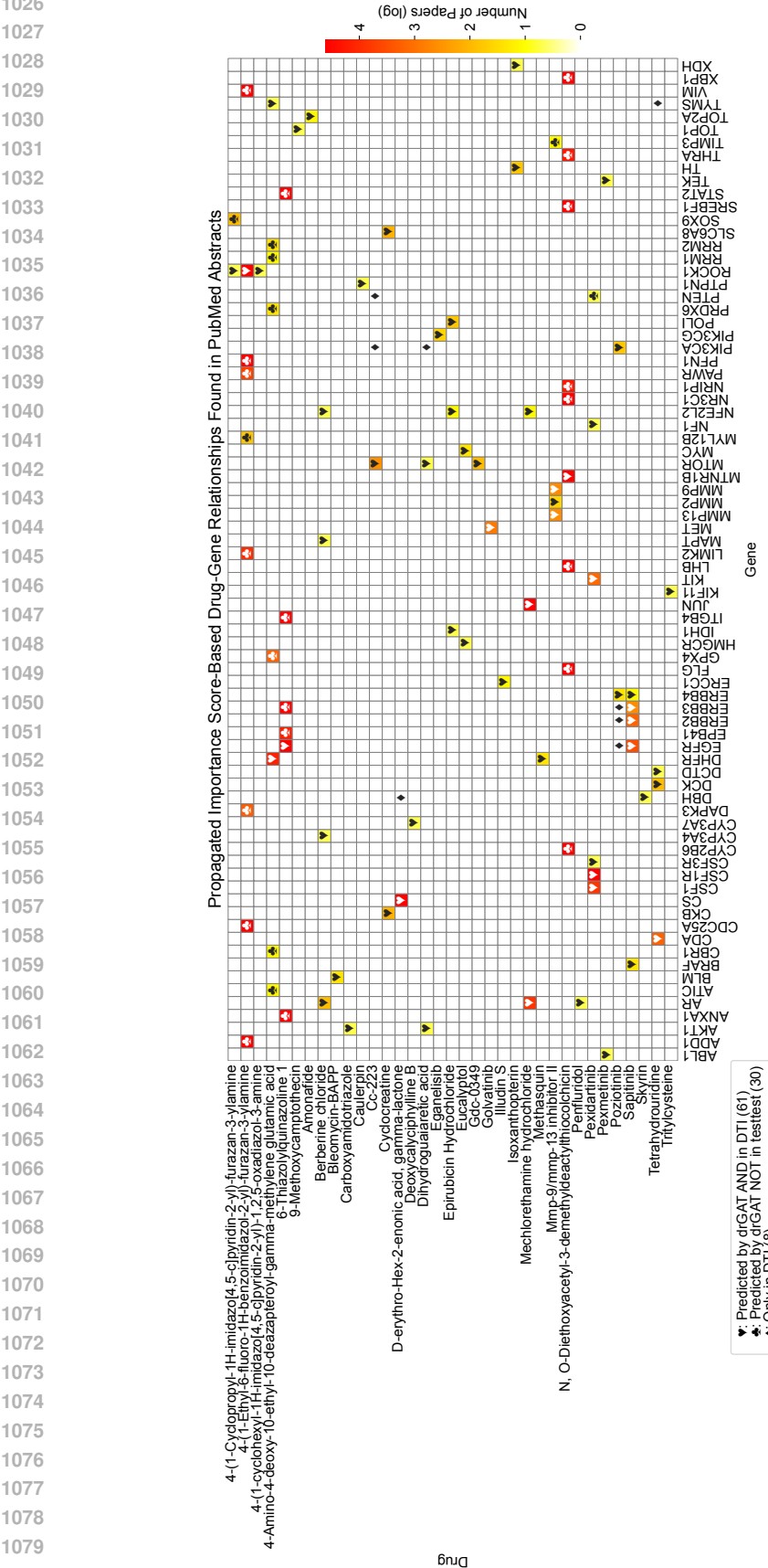

**Figure 4:** Selected drug-gene co-occurrences based on PubMed abstracts. The color represents the number of abstracts associated with a specific drug and gene pair by natural log scale. Symbols indicate the following: ♥ represents relationships predicted by GraphPINE and present in the Drug-Target Interaction (DTI) dataset (61 instances); ♣ represents relationships predicted by GraphPINE but not present in the DTI dataset (30 instances); ♦ represents relationships only present in the DTI dataset (8 instances). This figure shows a subset of the data for clarity

For non-zero DTI entries specifically, 90.5% of genes changed ranks with an average shift of ±69.71 (maximum: +946, minimum: -932). This demonstrates that our model modifies rankings for both zero and non-zero entries while preserving cosine similarity and rank correlation.

