# OpenReview forum: "GraphPINE: Graph importance propagation Neural Network for interpretable drug response prediction"
_ICLR.cc/2025/Conference — Submitted to ICLR 2025_

### Official Review · Reviewer_RCqp · 2024-10-25

**Soundness:** 3
**Presentation:** 3
**Contribution:** 2
**Rating:** 5
**Confidence:** 4

**Summary:**

This work proposes GraphPINE to handle the problem of drug response prediction. GraphPINE uses a graph neural network incorporating biomedical prior knowledge from various resources. As for the model architecture, GraphPINE utilizes Graph Transformer and GAT to handle features, and designs Importance Propagation Layer to provide understandings for nodes and their relations. In experiments, GraphPINE outperforms other baseline methods and provide interpretable results for drug response prediction examples.

**Strengths:**

1. This work utilizes different kinds of biomedical information to tackle drug response prediction, including gene-gene network, drug-target interaction and methylation.
2. GraphPINE introduces Importance Propagation Layer, which is good at processing information from multi sources.

**Weaknesses:**

1. This work lacks a problem formulation for the problem of drug response prediction, which makes the input and the output of the learning problem unclear.
2. The model architecture design is generally lack of novelty. For the feature processing part, Graph Transformer and GAT are existing works widely used. While the Importance Propagation Layer is similar to LSTM/GRU gates.
3. In the experiment part, all baselines are learning methods proposed several years ago and none of them is specially designed for drug response prediction. This makes the experimental comparison unreasonable, as the drug response prediction methods introduced in related work part are not compared.

**Questions:**

1. What are the differences between drug response prediction and other drug-related prediction problems (e.g. drug-target interaction prediction and drug-drug interaction prediction)?
2. In section 2.2 “graph neural network in computational biology”, why molecular property prediction methods are not mentioned? GNN is widely used for molecular property prediction problems.
3. Why the title of section 3.3.2 is “GraphPINE model”? It seems that the title and contents do not match.
4. In Figure 2, 3, 4, why there are color bars on the right? There are only two types of colors for nodes in the figure.

---

> ### Author Response · Authors · 2024-11-27
> **Comment to Reviewer RCqp**
>
> We thank the reviewer for their comments and response below:
>
> 1. We added problem formulation in L93-99. Drug response prediction is described y = f(G, D) where y is a prediction (e.g., IC50 or binary; drug sensitive or resistance) G is -omics data features and D is drug information.
> 2. Reviewer 1 shared similar concerns; please see our first comment in response to Reviewer 1 (kMoq).
> 3. We expanded our baseline comparisons in Table 1 (L435-446) to include additional state-of-the-art methods.
>
> Q1 Drug response prediction focuses on forecasting the overall biological response of cells or patients to drugs to predict metrics like IC50 values. Drug-target interaction prediction specifically examines molecular-level binding between drugs and protein targets. Drug-drug interaction prediction analyzes how multiple drugs affect each other's behavior, considering both pharmacokinetic and pharmacodynamic interactions to prevent adverse effects; (this was not covered in this study) (L44-50).
>
> Q2 We have now added molecular property prediction as an application of GNNs in computational biology in Section 2.2. Specifically, we included the sentence: "GNNs have also been used for molecule property prediction, showcasing the potential of GNNs in cheminformatics (Fu et al., 2021)" (L112-113).
>
> Q3 We revised the section title to "Model Architecture" since this section describes the overall architecture.
>
> Q4  You are correct that the color bars on the right of Figures 2, 3, and 4 are unnecessary since we only use two colors to distinguish node types in the figures. We removed these color bars in the revised version to avoid confusion and improve the clarity of the visualizations.

---

> > ### Comment · Reviewer_RCqp · 2024-11-27
> > **Thanks**
> >
> > Thank you for your answers in detail! Since my concerns are well handled, I have raised the overall score for the paper.

---

### Official Review · Reviewer_1Hic · 2024-11-02

**Soundness:** 3
**Presentation:** 4
**Contribution:** 3
**Rating:** 6
**Confidence:** 4

**Summary:**

The authors present GraphPINE, an interpretable GNN for drug response prediction. This methodology is able to deliver accurate results in terms of drug response prediction and, at the same time, provide interpretable outcomes in terms of important genes in the input graph. The authors properly described their methodology and compared their results with other methods. Finally, they show how it is possible to obtain and visualize interpretations for the predictions obtained.

**Strengths:**

The main strengths of the paper are the following:

1) The work is original and relevant since it touches on an important aspect of GNNs (interpretability). Instead of relying on external explainers (which can deliver biased results), the authors propose a way to render GNN interpretable by using importance score propagation.

2) The domain of application is of extreme importance and relevance since it could help facilitate the detection of drug response/resistance, speeding up drug development and clinical trial phases.

3) The interpretability results are in line with the knowledge present in the literature.

4) Overall, the work is well-presented and also the Appendix provides useful information.

5) The authors provided an anonymous repository for reproducibility

**Weaknesses:**

The main weaknesses of the paper are the following:

1) It is not clear to me how the initial importance scores are assigned. The authors say these scores are obtained using the weights of the edges of a knowledge graph. How is this done? My doubt is that if we start from consolidated importance scores, the result's final propagated importance will present a bias since it will be strongly dependent on the initial weights.

2) It is not clear to me if the authors compared their results against GAT, GT, and GINE. They present three GraphPINE versions based on those architectures, but a direct comparison with them is not provided. I am puzzled since, in the Appendix, they describe the hyperparameter tuning for those models, but no result from them is present in Table 1.

3) The improvement brought by GraphPINE in terms of evaluation metrics is marginal with respect to other methods. In particular, it would be interesting to see how GAT, GT, and GINE perform when used as standalone techniques.

**Questions:**

My questions are related to the weak points I described.

1) Can the authors better describe how the initial importance scores are obtained? This should be carefully described in the main paper and not in the Appendix.

2) If one used the initial importance scores to build a ranking of important nodes, would the outcome be different? If yes, then the methodology is effective; if no, probably the resutls are strongly affected by the initial importance scores.

3) Can the authors show how GAT, GT, and GINE perform when used as standalone strategies?

---

> ### Author Response · Authors · 2024-11-27
> **Comment to Reviewer 1Hic**
>
> We thank the reviewer; here are our responses: :
>
> 1. We moved the DTI data preprocessing steps to L212-215, L161-272; see Q2 for a response related to bias.
> 2. We added ablation studies (w/o IP layer) to Table 1.
> 3. We added ablation studies (w/o IP layer) to Table 1 and the comparison between with and without IP layer is explained in L439-446.
>
> Q1 We have expanded the methodology section (L212-215, L261-272) to detail our DTI score calculation process. The scores are normalized counts of PubMed article co-mentions.
>
> Q2 We have added the section “Evaluation of Importance Score Propagation” (L480-503) to further discuss how our DTI importance values shift as a result of propagation.
>
> Q3 Table 1 now includes ablation studies. GraphPINE consistently outperforms them by 2-13% across all metrics except the specificity of Graph Transformers.

---

> > ### Comment · Reviewer_1Hic · 2024-11-27
> > **Response to Authors**
> >
> > Thank you for addressing my concerns. The paper's quality slightly improved, but I kept my score as is. Despite being a good paper, I do not think the content is enough to raise the score to 8.

---

### Official Review · Reviewer_kMoq · 2024-11-04

**Soundness:** 2
**Presentation:** 2
**Contribution:** 2
**Rating:** 5
**Confidence:** 4

**Summary:**

The manuscript introduces GraphPINE, a Graph Neural Network (GNN) architecture aimed at enhancing interpretable drug response prediction. By leveraging prior biological knowledge through a knowledge graph with weighted edges, GraphPINE generates initial importance scores for nodes. The core innovation of the work is the Importance Propagation (IP) layer, which facilitates the propagation of node importance throughout the GNN, thereby promoting biological interpretability.

**Strengths:**

1. The manuscript addresses an important challenge in drug response prediction, highlighting the need for interpretable models in biomedical applications.

2. The integration of a knowledge graph to inform initial importance scores is a compelling approach that could enhance the interpretability of GNNs in drug response contexts.

**Weaknesses:**

1. The methodology lacks novelty, as graph convolutional networks (GCN) and importance gating have been previously employed in similar contexts. The claim of novelty in this paper is undermined by the existence of other interpretable GNN-based methods for drug response prediction.

2. The ablation study is missing, and important baseline models are not included in the comparisons. The results show only marginal improvements over baseline models. For instance, ROC-AUC results show GraphPINE at 0.7955 compared to LightGBM at 0.7901, and PR-AUC results indicate 0.8939 vs 0.8917. These limited improvements weaken the overall impact of the claims.

3. The authors should provide comprehensive comparative analyses with and without drug-target interaction (DTI) information to demonstrate the importance of DTI. If predictive performance is comparable to models that exclude DTI, this raises questions about the effective utilization of the interaction information.

**Questions:**

see above.

---

> ### Author Response · Authors · 2024-11-27
> **Comment to Reviewer kMoq**
>
> We thank the reviewer address concerns below:
>
> 1. We clarified our contributions in the Abstract (L18-29) and in a new sub-section (Important Gating with GNNs, L166-186) that compares our approach with existing methods and describes our unique architectural improvements. To summarize, GraphPINE leverages domain-specific prior knowledge for node importance score initialization. Use cases in biomedicine necessitate generating hypotheses related to specific nodes. Commonly, there is a manual post-prediction step examining literature (i.e., prior knowledge) to better understand features. While node importance can be obtained for gradient and attention-based methods after prediction, these node importances lack complementary prior knowledge; GraphPINE seeks to overcome this limitation. GraphPINE differs from other GNNs with gating methods that utilize an LSTM-like sequential format such that we introduce an importance propagation layer that unifies 1) updates for feature matrix and node importances, jointly and 2) uses GNN-based graph propagation of feature values; update Figure 1A to be more explicit.
> 2. Regarding empirical validation and performance gains: We conducted ablation studies showing the IP layer impact (incorporating DTI information). We expanded our baseline comparisons in Table 1 (L435-446) to include additional methods. GraphPINE consistently outperforms them by 2-13% across all metrics except the specificity of Graph Transformers (Table 1). The results confirm that incorporating DTI information through the IP layer consistently improves prediction performance across multiple metrics. While the ROC-AUC/PR-AUC improvements may be considered minor, the result is then a technically equivalent model designed with interpretability constraints in mind. These constraints align with model result usage, in practice, as hypothesis generators for decision-making of wet-lab biology experiments where the reasonability of any predictor is first weighed against its biological function and other literature sources.
> 3. The DTI information is directly tied to the IP layer in GraphPINE. Therefore, please check the above answer about the with/without DTI information experiments.

---

### Author Response · Authors · 2024-11-27
**General Response**

We thank all reviewers for their review and constructive feedback. In this general response, we summarize our main updates and the additional experiments we performed. In the updated paper we highlight major changes in blue.

## Updates to our main results

We have made important updates to our main results that are detailed in reviewer responses.
1. Added ablation study and drug response prediction baseline methods.
2. Clarify the novelty of GNN with gating methods comparison with previous methods.
3. Evaluate the importance propagation between the initial and propagated score.

#### Minor updates
- We explain more the use case being examined (i.e., the interplay between drug response and drug target interaction prediction)
- We add more related works
- We move details about generating our DTI dataset from appendix to main text
- We edit section titles for clarity

---

### Meta-Review · Area_Chair_jyrq · 2024-12-17

**Metareview:**

The paper addresses drug response prediction using a novel GNN-based model, GraphPINE. Reviewers raised concerns about the novelty of the approach, as the methods employed (e.g., GCN, importance gating, and GAT) are well-established. The lack of an ablation study, insufficient baseline comparisons, and marginal improvements over existing models were also highlighted. The authors have clarified the novelty of their work, emphasizing the unique integration of domain-specific prior knowledge and importance propagation. They also addressed the absence of molecular property prediction methods and expanded experimental comparisons. Despite these clarifications, the overall contributions lack substantial innovation and the experimental validation is not convincing enough to justify publication. The paper’s incremental advances and limited empirical support led to the decision for rejection.

**Additional Comments On Reviewer Discussion:**

While the authors have provided responses, they have not adequately addressed the related concerns. No reviewers want to champion this paper for an acceptance.

---

### Decision · Program_Chairs · 2025-01-22

Reject